# Experimental Study on the Skyhook Control of a Magnetorheological Torsional Vibration Damper

**DOI:** 10.3390/mi15020236

**Published:** 2024-02-02

**Authors:** Zhicheng Wang, Hongsheng Hu, Jiabin Yang, Jiajia Zheng, Wei Zhao, Qing Ouyang

**Affiliations:** 1College of Mechanical Engineering, Zhejiang University of Technology, Hangzhou 310014, China; 2College of Information Science and Engineering, Jiaxing University, Jiaxing 314001, China; 3College of Engineering, Zhejiang Normal University, Jinhua 321004, China; 4School of Mechanical Engineering, Nanjing University of Science and Technology, Nanjing 210044, China; 5Taizhou Jiuju Technology Co., Ltd., Taizhou 318000, China

**Keywords:** magnetorheological dampers, torsional vibration, semi-active control, skyhook control

## Abstract

This study proposes a dual-coil magnetorheological torsional vibration damper (MRTVD) and verifies the effectiveness of semi-active damping control to suppress the shaft system’s torsional vibration via experimental research. Firstly, the mechanical model of the designed MRTVD and its coupling mechanical model with the rotating shaft system are established. Secondly, the torsional response of the shaft system is obtained via resonance experiments, and the influence of the current on the torsional characteristics of the magnetorheological torsional damper is analyzed. Finally, the MRTVD is controlled using the skyhook control approach. The experimental results demonstrate that when the main shaft passes through the critical speed range at various accelerations, the amplitude of the shaft’s torsional vibration decreases by more than 15%, and the amplitude of the shaft’s torsional angular acceleration decreases by more than 22%. These conclusions validate the inhibitory effect of MRTVD on the main shaft’s torsional vibrations under skyhook control.

## 1. Introduction

In the engine system, the crankshaft system is influenced by periodic inertial moments, leading to the occurrence of shaft torsional vibrations [1]. Especially in conditions involving substantial rotational inertia, high rotational speeds, and heavy loads [2], as well as uneven cylinder ignition, the crankshaft’s torsional vibrations become intensified [3]. Severe torsional vibrations can result in fatigue damage to the crankshaft [4], wear of meshing gears [5], the generation of harmful noise [6], and the incurrence of energy transmission losses [7]. To prevent crankshaft torsional deformation or failure, effective measures should be taken to dampen the torsional vibrations [8]. Currently, the primary method for attenuating crankshaft torsional vibrations is by installing a torsional vibration damper in the crankshaft system [9], which can effectively circumvent crankshaft torsional vibrations induced by resonance and reduce the amplitude. Such torsional vibration dampers can effectively control the crankshaft system, ensure its operation within a reasonable vibration range, and ultimately enhance the stability and reliability of the system.

Up to now, various torsional vibration dampers have been produced and deployed, including the silicon oil damper [10], rubber damper [11], and silicon oil–rubber composite torsional vibration damper [12]. However, these dampers only exhibit favorable damping effects at specific resonance frequencies, and their damping is uncontrollable [13]. To overcome these limitations, some researchers have proposed composite or hybrid damping approaches. Yu designed a torsion–bending compound rubber damper [14], which significantly reduces airborne noise. Sezgen and Tinkir employed a hybrid damping method to optimize damper design, demonstrating that the hybrid damping approach is more effective than a singular damping method [15]. Nevertheless, once the passive torsional damper’s design is completed, the dynamic parameters, including damping and stiffness, remain fixed [16]. This implies that deviations in manufacturing and installation techniques, temperature-dependent performance degradation, and other factors may lead to mismatched performance between the designed torsional damper and the crankshaft system [17].

For dealing with the problems of the aforementioned passive dampers, a more flexible and controllable torsional vibration damping approach needs to be sought. In this regard, the magnetorheological fluid (MRF) smart material-based damper emerges as an excellent choice. MRF dampers possess adjustable damping, a wide dynamic range, rapid responsiveness, and low power consumption, making them extensively employed in various intelligent vibration suppression applications. Numerous scholars have conducted relevant research and exploration on magnetorheological torsional vibration dampers. Ye and Williams investigated torsional vibration control using a magnetorheological brake, significantly reducing the vibration amplitude of the main system by controlling the magnetorheological brake [18]. Abouobaia et al. designed a rotating disc-type magnetorheological torsional vibration damper to attenuate shaft torsional vibrations and validated the model’s optimization strategy by testing damping torque under various electrical current excitations [19]. Subsequently, they developed a novel hybrid torsional damper by combining a magnetorheological damper with a centrifugal pendulum damper. Furthermore, simulation comparisons between passive control and skyhook control algorithms for torsional vibration control confirmed the effectiveness of semi-active control [20]. He et al. proposed a semi-active skyhook control method for magnetorheological torsional dampers and conducted joint control simulation analysis to verify the effectiveness of the control strategy [21]. Yang and Guo designed a control system for multiple MR dampers, and via modal analysis and nonlinear time history analysis, the effectiveness of the MR control system was demonstrated, significantly attenuating the torsional vibrations of eccentric structures [22]. Liu et al. provided a convenient optimization design method for magnetorheological torsional vibration absorbers by analyzing axial single-coil configuration, axial multi-coil configuration, and circumferential configuration [23]. Li et al. developed a novel magnetorheological torsional vibration absorber with variable stiffness and damping, employing independent damping control and independent stiffness control to suppress the torsional vibrations of transmission systems, thus validating the effectiveness of semi-active control in attenuating torsional vibrations in transmission systems [24]. Gao et al. proposed a control strategy combining transient lookup table and steady-state optimization using magnetorheological elastomers to significantly reduce deviations from identified dominant frequencies of external excitations, enabling the absorber to rapidly track the dominant external excitation frequency, thereby achieving semi-active control absorption of vibrations [25]. However, there is still a lack of extensive research on the damping performance of magnetorheological torsional vibration dampers under experimental conditions and their semi-active control of shaft torsional vibration in the literature. Therefore, designing a rational magnetorheological torsional vibration damper and investigating its semi-active control effect on shaft torsional vibration are crucial for comprehensively understanding its damping performance under experimental conditions and resolving shaft torsional vibration issues.

Against this backdrop, a dual-coil magnetorheological torsional vibration damper (MRTVD) is proposed based on the rheological effect of MRF. Firstly, the structure of MRTVD is designed, and the underlying principle of the designed MRTVD is analyzed. Secondly, its dynamic model is derived, and a coupled model of MRTVD and the shaft system is established based on the torsional vibration experimental setup of the shaft system. Finally, the torsional response of the shaft system is obtained via resonance experiments, ensuring effective vibration reduction in the experimental environment and enabling semi-active control of the shaft system’s torsional vibration.

## 2. Structure and Dynamic Model of MRTVD

### 2.1. MRTVD Structural Design

As shown in Figure 1, the MRTVD consists of an upper cover plate, a shell body, an inertia body, a lower cover plate, excitation coils, and a slip ring. Two excitation coils are fixed at the ends of the outer shell, while the gap between the outer shell and the inertia ring is filled with MRF. The MRTVD is designed in shear mode, which features minimal zero-field damping force, a simple structure, and easy assembly.

When the external power supply is turned off, no magnetic field is generated within the damper; in this case, the MRF exhibits zero-field viscosity, and the MRTVD is a passive-like damper. When the external power supply is turned on, the current flows through the conductive slip ring through the two sets of coils to form a closed loop, and the induced magnetic field is generated. These magnetic induction lines pass through the inertia body and the end face damping channel, and then return to the inertia body through the cover plate, shell body, and shaft face damping channel, creating a closed magnetic field. In this scenario, the flow of the MRF in the damping channel is perpendicular to the magnetic lines of force. Due to the influence of the magnetic field, the magnetorheological effect occurs, and simultaneous relative motion between the inertia block and the shell results in a shear working mode, generating output damping force to absorb torsional vibration energy and achieve torsional vibration reduction.

The MRF used in this study is the medium-density type MRF-350. The magnetization characteristics and mechanical properties of this MRF are illustrated in Figure 2. The approximate mathematical relationship between magnetic flux density and shear stress of MRF-350 can be represented by the polynomial fitting method as follows:(1)$$\tau_{B}=-74554B^{4}+68848B^{3}+65619B^{2}+17637B-153.6913$$

The nearly linear relationship between the magnetic field intensity H and the magnetic induction intensity B (from 0.2 T to 1.1 T) can be seen in Figure 2. The relative permeability of MRF-350 is approximately 4.77. The shear stress increases rapidly as the magnetic induction strength increases, and the growth trend gradually tends to be flat once the magnetic induction strength reaches 0.8 T. Table 1 presents the performance of the MRF-350 material under a shear rate of 10 (1/s).

Firstly, the model of the proposed MRTVD was developed using Solidworks software 2020, as depicted in Figure 3a, with the provided parameters listed in Table 2. A two-dimensional axisymmetric model was established in MAXWELL, with simulation settings illustrated in Figure 3b. The model incorporated relevant material parameters and current excitation, enabling the determination of the magnetic field distribution within the structural damper (depicted in Figure 3c,d), as well as the magnetic field intensity within the fluid gap (illustrated in Figure 3e,f).

The curve in Figure 3c shows that in the fluid gap at the end face, the change in magnetic field intensity is approximately linear, and the magnetic field intensity is greater near the coil. This is because the closer to the coil, the shorter the magnetic field loop formed, resulting in a smaller magnetic resistance in the magnetic circuit. However, as the radius changes, the proportion of magnetic resistance involved in the entire magnetic circuit is relatively small, resulting in a more gradual change in magnetic field intensity. The curve in Figure 3d depicts that in the circumferential liquid gap, the magnetic field intensity in the magnetostrictive region is extremely high and evenly distributed, but there are fluctuations in the edge region. The magnetic field in the non-magnetic field areas on both sides is close to 0 mT, while in the edge area, the magnetic field strength will rapidly increase.

### 2.2. Working Principle of MRTVD in Shaft System

The analysis of shaft torsional vibration often involves simplifying the entire shaft system into an equivalent undamped single-degree-of-freedom system [13]. With the installation of a torsional vibration damper at the free end of the shaft system, the entire system can be further simplified into a forced two-degree-of-freedom vibration system, as illustrated in Figure 4.

The dynamic equations of the two-degree-of-freedom torsional vibration model are established as follows:(2)$$\left\{\begin{array}{c} J_{g}{\ddot{\theta }}_{g}+C_{d}({\dot{\theta }}_{g}-{\dot{\theta }}_{d})+K_{d}(\theta_{g}-\theta_{d})+K_{g}\theta_{g}=M(t)\\ J_{d}{\ddot{\theta }}_{d}-C_{d}({\dot{\theta }}_{g}-{\dot{\theta }}_{d})-K_{d}(\theta_{g}-\theta_{d})=0 \end{array}\right.$$
where $J_{g}$ and $K_{g}$ are the rotational inertia and stiffness of the main system; $J_{d}$, $K_{d}$, and $C_{d}$ is the rotational inertia, torsional stiffness, and viscous damping coefficient of the torsion damper; $\theta_{g}$ is the torsional vibration angle of the main system; $\theta_{d}$ is the torsional vibration angle of the magnetorheological torsion damper; and M(t) is the excitation torque.

Given the excitation torque $M(t)=Me^{iwt}$, and assuming the form of the solution as $\theta_{g}=A_{g}e^{iwt}$ and $\theta_{d}=A_{d}e^{iwt}$, the amplitude $A_{g}$ of the main system is expressed as follows:(3)$$A_{g}=\frac{M(K_{d}-J_{d}\omega^{2}+iC_{d}\omega )}{\left[(K_{g}-J_{g}\omega^{2})(K_{d}-J_{d}\omega^{2})-J_{d}K_{d}\omega^{2}]+iC_{d}\omega (K_{g}-J_{g}\omega^{2}-J_{d}\omega^{2}\right)}$$

Let $\mu =\frac{J_{d}}{J_{g}}$ be the inertia ratio between the torsional vibration damper and the main system; $\lambda =\frac{\omega }{\omega_{g}}$ represents the frequency ratio of forced vibration; $\gamma =\frac{\omega_{d}}{\omega_{g}}$ represents the natural frequency ratio between the torsional vibration damper and the main system; $A_{st}=\frac{M}{K_{g}}$ represents the static deflection of the main system; and $\zeta =\frac{C_{d}}{2\sqrt{J_{d}K_{d}}}$ is the damping ratio. The amplification factor of the shaft torsional vibration is expressed as follows:(4)$$R_{g}=|\frac{A_{g}}{A_{st}}|=\sqrt{\frac{{\left(\gamma^{2}-\lambda^{2}\right)}^{2}+{\left(2\zeta \lambda \right)}^{2}}{{\left(2\zeta \lambda \right)}^{2}{\left(\lambda^{2}-1+\mu \lambda^{2}\right)}^{2}+{\left[\mu \lambda^{2}\gamma^{2}-(\lambda^{2}-1)(\lambda^{2}-\gamma^{2})\right]}^{2}}}$$

As a typical two-degree-of-freedom system, variations in the damping ratio ζ of the damper result in different amplitude curves of the system, as shown in Figure 5.

When the damping ratio varies within the range of [0, 1] and the frequency ratio varies within the range of [0, 2], the surface plot of the amplification factor of shaft torsional vibration is depicted in Figure 6. It can be seen that both the damping ratio and frequency ratio of the torsional vibration damper significantly influence the multiplication factor of the shaft system. At any given excitation frequency, there exists an optimal dynamic damping value that minimizes the torsional vibration amplitude of the main system. In other words, for different excitation frequencies, selecting the corresponding dynamic optimal damping (illustrated by the ideal control curve in Figure 5) allows for the suppression of torsional vibration in the main system. This provides a crucial theoretical foundation for subsequent variable damping torsional vibration control.

## 3. Mechanical Modeling of MRTVD and Shaft System

### 3.1. Dynamic Model of MRTVD

Its output damping torque $T_{D}$ is primarily composed of two parts [23]. One part is the damping torque $T_{\tau }$ generated by the MRF in the torsional vibration damper when undergoing the rheological effect. The other part is the damping torque $T_{\eta }$ produced by the inherent viscosity of the MRF under zero-field conditions. Therefore, the output torque of the magnetorheological torsional vibration damper can be expressed as follows:(5)$$T_{D}=T_{\tau }+T_{\eta }$$

Based on Newton’s law of internal friction, the equivalent yield stress generated by the MRF in the fluid gap can be expressed as
(6)$$\tau =\frac{F}{A}=\eta \dot{\gamma }=\eta \frac{\nu }{\delta }$$

The Bingham model of MRF can be represented as [26]
(7)$$\tau =\tau_{H}+\eta \dot{\gamma }$$
where η represents the fluid viscosity; ν denotes the relative velocity between the polar plates; δ represents the distance between the polar plates; $\tau_{H}$ represents the yield stress; and $\dot{\gamma }$ stands for the fluid shear rate.

As shown in Figure 7a,b, the rotational damper possesses two working regions: the circumferential and end faces. The damping torques acting on the circumferential and end faces of the damper can be expressed as follows:(8)$$M_{z}=(\tau_{B}+\eta_{0}\frac{\omega_{r}R}{\delta })A_{B}R+(\tau_{0}+\eta_{0}\frac{\omega_{r}R}{\delta })A_{0}R$$
(9)$$M_{d}=2[\int_{R_{1}}^{R}(\tau_{B}+\eta_{0}\frac{\omega_{r}R}{\delta })A_{B}RdR+\int_{R_{1}}^{R}\left(\tau_{0}+\eta_{0}\frac{\omega_{r}R}{\delta })A_{0}RdR\right]$$
where $\tau_{B}$ and $\tau_{0}$ represent the shear yield stresses of the MRF under magnetic field and non-magnetic field conditions, respectively. $\eta_{0}$ denotes the yield viscosity of the MRF under zero-field conditions. $w_{r}$ is the speed difference between the inertia ring and the shell. $A_{B}\;$and $A_{0}$ represent the areas of the regions with and without a magnetic field in the fluid gap, respectively. R is the rotational radius.

The output damping torque of MRTVD can be expressed as follows:(10)$$M=\frac{4\pi \tau_{B}}{3}(R_{1}^{3}-R_{n}^{3})+\frac{\pi \eta_{o}w_{r}}{\delta }(R_{1}^{4}-R_{n}^{4})+2\pi R_{1}^{2}C_{1}(\tau_{0}+\eta_{0}\frac{\omega_{r}R_{1}}{\delta })+2\pi R_{1}^{2}C_{2}(\tau_{B}+\eta_{0}\frac{\omega_{r}R_{1}}{\delta })$$
where $R_{1}$ represents the outer radius of the working gap for the MRF. $C_{1}$ denotes the working gap width of the MR fluid in the circumferential direction under zero-field conditions. $C_{2}$ stands for the working gap width of the MR fluid in the circumferential magnetic field.

### 3.2. Torsional Vibration Mechanical Model of MRTVD-Based Shaft System

Investigating the effect of MRTVD on controlling shaft torsional vibration, a structural configuration of the shaft torsional vibration system was designed, as depicted in Figure 8.

The rotational motion of the shaft torsional vibration system is achieved via the composition of the servo motor, synchronous pulley drive, MRTVD, and the main shaft. On the test bench, an adjustable-frequency continuous torsional excitation torque is generated by the servo motor and the exciter. The main shaft speed signal is acquired using two encoders, and a series of signal analysis processes are conducted to obtain the torsional vibration angle of the main shaft. The adjacent components of the torsional vibration system are connected using shafts and couplings with specified stiffness. The transmission system is supported by bearing seats and structural frames, which are fixed to the external support of the vibration table, forming the actual structure of the transmission system.

Figure 9 displays the basic structure and equivalent torsional vibration model of the shaft torsional vibration system experimental setup. The experimental setup comprises the torsional excitation system, the main shaft, and the MRTVD. Rotational motion and the application of excitation torque are accomplished by the torsional excitation system, while the shell body of the MRTVD is rigidly connected to the main shaft. The rotational inertias of the torsional excitation system, the damper shell body, and the damper inertial body are denoted as $J_{1}$, $J_{2}$, and $J_{3}$, respectively, with corresponding rotation angles represented as $\theta_{1}$, $\theta_{2}$, and $\theta_{3}$. An elastic coupling, simplified as torsional stiffness $K_{0}$, connects the torsional excitation system to the MRTVD’s shell body through the main shaft. The damper shell body and the inertial body are connected by an MRF with a variable damping coefficient C, where M denotes the amplitude of the excitation torque applied to $J_{1}$, and w denotes the excitation frequency.

The rotational inertia of MRTVD components and the torsional stiffness of the shaft are calculated by software, as shown in Table 3.

Given that there is no elastic connection between the inertia block $J_{3}$ and the excitation system $J_{2}$ in the magnetorheological torsional vibration damper, the system can be simplified as a double torsion pendulum for analysis. Consequently, the equation of motion is simplified to
(11)$$\left\{\begin{array}{c} J_{3}{\ddot{\theta }}_{3}+C\left({\dot{\theta }}_{3}-{\dot{\theta }}_{2}\right)=0\\ J_{2}{\ddot{\theta }}_{2}-C({\dot{\theta }}_{3}-{\dot{\theta }}_{2})+K_{0}\theta_{2}=M(t) \end{array}\right.$$

Solve to obtain the following:(12)$$\Delta =\alpha +ib=I_{2}I_{3}\omega^{4}-(I_{2}+I_{3})K_{0}\omega^{2}+C\omega [K_{0}-\omega^{2}(I_{2}+I_{3})]i$$
(13)$$\Delta_{21}=c+id=0+Cwi$$
(14)$$\Delta_{22}=e+if=-\omega^{2}I_{3}+C\omega i$$

The amplitudes of the two mass points at this point can be expressed as
(15)$$A_{1}=M_{2}\sqrt{\frac{c^{2}+d^{2}}{a^{2}+b^{2}}}\;A_{2}=M_{2}\sqrt{\frac{e^{2}+f^{2}}{a^{2}+b^{2}}}$$

In the above equation, assuming the existence of $\frac{b^{2}}{a^{2}}=\frac{f^{2}}{e^{2}}$ then ±af=be, $A_{2}=M_{2}\frac{e}{a}$ is independent of the damping C. Organizing the equation gives
(16)$$\omega^{4}-\frac{2}{2+\mu }\omega_{n}^{2}\omega^{2}=0$$
(17)$$A_{2}=M_{2}\frac{1}{K_{0}-{\omega_{M}}^{2}(I_{2}+I_{3})}$$

By performing partial differentiation and selecting the optimal damping ratio, denoted as $\gamma =\frac{c}{2I_{3}\omega_{n}}$, and then expressing the results in terms of amplification factors, the equation can be simplified as follows:(18)$$\gamma =\frac{C}{2I_{3}\omega_{n}}=\frac{1}{\sqrt{2(1+\mu )(2+\mu )}}$$

During the operation of the shaft system, optimal damping effects of the MRTVD can be achieved by adjusting the excitation current of its coils based on the system’s various resonance frequencies and different operating speeds of the main shaft. The real-time adjustment of the excitation current ensures that the system remains in its optimal damping state, thus effectively controlling the system’s torsional vibration. This adaptive control strategy allows the MRTVD to flexibly address torsional vibration issues under different operating conditions, optimize damping performance, and enhance the system’s stability and reliability.

## 4. Experimental Results and Discussion

To accurately determine the system’s resonance frequency, the excitation torque and frequency amplitude of the exciter are set, the shaft is then uniformly accelerated from a stationary state until reaching a certain velocity, and then moved at a constant speed. This approach facilitated the successful acquisition of velocity data at both ends of the shaft, as demonstrated in Figure 10. The adoption of this method ensured a high degree of accuracy and enhanced the reliability and reproducibility of the experimental results. Figure 10 presents the impact of frequency variation on the angular velocity at both ends of the shaft, effectively emphasizing the crucial characteristics of the system’s resonance point.

Based on the relationship between the actual speed and time at both ends of the spindle, as shown in Figure 10A, a significant speed fluctuation is observed within 1–3 s. Subsequently, via data processing and analysis, the variation in the system’s torsional vibration angle was obtained. FFT analysis was conducted on the angle variation within 1–3 s, yielding the results shown in Figure 10B. It is noteworthy that a prominent peak appears at 50.89 Hz on the Fourier analysis curve. Therefore, the resonance frequency of the experimental platform is in the vicinity of 50.89 Hz. The average resonance frequency of the MRTVD torsional vibration test bench was measured to be 50.58 Hz, with a standard deviation of 0.37, via multiple parallel experiments.

### 4.1. Variable Damping Control Experiment

By installing MRTVD on the shaft system, the damping of the main shaft’s torsional vibration can be controlled via current adjustment. When 0 A and 1 A currents were applied to the MRTVD excitation coil, the torsional vibration signal of the main shaft at resonance was observed, as illustrated in Figure 11a. To gain a clearer understanding of the torsional vibration variations in the main shaft in resonance, the angular acceleration fluctuations of MRTVD were extracted at different currents and represented as curves in Figure 11b. The results of the time-domain analysis demonstrate a significant reduction in main shaft speed fluctuation and angular acceleration fluctuation when the current is set at 1 A. This reduction indicates a noticeable attenuation of torsional vibrations in the shaft system under resonance conditions, validating the effectiveness and damping performance of the vibration damper.

To investigate the vibration reduction effect of the designed MRTVD, the main shaft was intentionally set to resonate at a constant speed with an excitation frequency of 50.89 Hz. The speed fluctuations of the main shaft were then studied during uniform motion in the vicinity of the resonance point, and subsequent analyses were carried out. The MRTVD excitation coils were subjected to varying currents (ranging from 0 A to 2 A in increments of 0.25 A) to explore their influence on the main shaft’s speed fluctuations. The resulting experimental data, depicted in Figure 12, illustrate the changes in torsional angle amplitude during resonance under different current conditions.

The variable damping characteristics of the operating shaft system are unequivocally and significantly improved by the utilization of MRTVD, as demonstrated by the experimental results. MRTVD exhibits remarkable torsional vibration suppression performance via the implementation of appropriate current control. Under resonance conditions, a reduction of 30% in the amplitude of the torsional angle of the main shaft was observed at the optimal current, achieved by applying different currents to the MRTVD excitation coils during shaft operation. This observation underscores the potential to utilize the controller for inputting the optimal current to the MRTVD during torsional vibration, thereby effectively mitigating the torsional vibrations within the shaft system.

As shown in Figure 5, it can be observed that after installing the torsional vibration absorber on the shaft system, the amplitude of the shaft system will also experience fluctuations during the damping variation in the absorber. As the damping in the torsional vibration absorber gradually increases, the friction between the shell and the inertia body on the shaft system also increases. Therefore, as the damping of the absorber changes, its frequency also varies. During this damping variation process, when the damping reaches a certain value, the relative rotation between the shell and the inertia body of the absorber reaches its maximum, effectively reducing the vibration energy generated by the torsional vibration of the shaft system to the maximum extent. This point is the optimal damping point, and the current at this point is the optimal current. At the optimal damping, the absorber exhibits the best vibration reduction effect.

### 4.2. Semi-Active Control Experiment

The skyhook damping control strategy assumes a pivotal role in semi-active suspension control, standing out as the most widely employed approach in magnetorheological semi-active control. This strategy is characterized by its simplicity, ease of implementation, and robust performance. This study aims to validate the practical control efficacy of the MRTVD via the application of the skyhook control strategy. According to the skyhook control principle, when the product of velocity fluctuations and acceleration is greater than zero, indicating the forward motion of the shaft system, the ideal skyhook damper should exert a force on the main shaft opposite to the direction of the shaft system’s motion to counteract torsional vibration. Consequently, the output force of MRTVD should be equal to that of the ideal skyhook damper. Conversely, when the product of velocity fluctuations and acceleration is less than zero, MRTVD should output the minimum damping force, corresponding to minimal current control. The control strategy of the skyhook control is presented for practical and simplified purposes as follows [27]:(19)$$\begin{array}{cc} c=c_{max} & if\;e\dot{e}\geq 0\\ c=c_{min} & if\;e\dot{e}<0 \end{array}$$
where c is the damping, $c_{max}$ is the maximum damping factor, $c_{min}$ is the minimum damping factor, e is the angular velocity, and $\dot{e}$ is the angular acceleration.

At this juncture, the current is set at either 0 A or adjusted to the optimal control current for the current resonance speed. The control process of shaft torsional vibration involves the real-time acquisition of vibration signals of the shaft. These signals are then analyzed and calculated by the controller to generate the appropriate corresponding currents. Subsequently, the current is input to the MRTVD to produce the corresponding damping force, thereby achieving effective vibration reduction.

The mitigation of shaft torsional vibration is accomplished via a controlled process involving the acquisition of real-time vibration signals from the shaft system. These signals are subsequently analyzed and processed by the controller, leading to the computation of relevant data. The controller then generates an appropriate current, which is utilized to modulate the MRTVD. This enables the MRTVD to produce a corresponding damping force, effectively reducing vibrations. Consequently, when employing skyhook control in the MRTVD system, the control system’s block diagram is illustrated in Figure 13.

Our actual control involves applying the input current magnitude to the MR damper. Under the skyhook control strategy, we simply set the current to either 0 or Imax, corresponding to zero-field damping and maximum control damping, respectively. By using combined simulation with Amesim software 16 and MATLAB 2016, the parameter equations were numerically solved to obtain the spindle’s vibration curve more intuitively. Based on this, simulation analysis was conducted on the angular velocity, acceleration, and torsional angle of the system under MRTVD damping control. Figure 14 illustrates the torsional vibration response of the spindle system under passive control (without MRTVD control) and skyhook control under acceleration conditions, showing the angular velocity, angular acceleration, and torsional angle.

From the simulation analysis graph, it is evident that skyhook control is significantly superior to passive control, effectively suppressing the angular velocity and angular acceleration amplitudes of the shaft system’s torsional vibration near the resonance frequency under acceleration conditions. Based on the simulation results, peak and root mean square values of shaft angle fluctuation, as well as peak and root mean square values of angular acceleration fluctuation, were computed, as shown in Table 4.

Based on the simulation results, as shown in Table 3, peak and root mean square values of shaft angle fluctuation, as well as peak and root mean square values of angular acceleration fluctuation, were computed. From the table above, it can be observed that, compared to passive control under acceleration conditions, skyhook control can reduce the angular fluctuation and angular acceleration fluctuation of the shaft system by 26.18% and 25.28%, respectively. The root mean square (RMS) values of angular fluctuation and angular acceleration fluctuation are reduced by 22.73% and 25.02%, respectively.

A comparison between skyhook control and passive control was conducted to assess their effects on the torsional vibration of the shaft system, as illustrated in Figure 15. The experimental results clearly demonstrate that skyhook control is significantly more effective than passive control near the resonance frequency, with successful suppression of the amplitudes of angular velocity and angular acceleration of the shaft system at the resonance point. Moreover, the torsional vibration response during the stable operating phase was also reduced to some extent by skyhook control.

Based on the experimental results mentioned above, the peak and root mean square values of torsional vibration fluctuations, along with the peak and root mean square values of angular acceleration in the resonant state of the shaft system, were calculated and are listed in Table 5. The numbers in parentheses indicate the percentage reduction in vibration response compared to the passive control case.

Table 5 clearly demonstrates that under the resonant condition, significant reductions of 19.8% and 20.1% in the root mean square values of both shaft torsional vibration fluctuations and angular acceleration fluctuations were achieved with skyhook control, compared to passive control. Additionally, the peak values were reduced by 16.9% and 28.4%, respectively. These analytical results provide comprehensive validation of the effectiveness of the skyhook’s semi-active control and emphasize its remarkable performance in vibration reduction.

Both simulation and experimental results validate that employing MRTVD with skyhook control can effectively and stably suppress shaft torsional vibrations, providing crucial assurance for the reliable operation of the spindle.

Via separate experiments, the rise of the main shaft’s torsional rotation from a state of rest to the operating speed of 840 rpm under various accelerations was measured for both the skyhook control and passive control strategies. Additionally, the inhibitory effect of the MRTVD on the main shaft’s torsional rotation was evaluated. To compare the effects on the torsional vibration of the shaft system, a specific acceleration value of $\alpha =15\pi \;rad/s^{2}$ was applied. The specific data obtained from this experiment are presented in Figure 16. The results indicate a significant reduction in the amplitudes of the main shaft’s torsional vibration fluctuations and angular acceleration fluctuations by 20.5% and 28.0%, respectively, under skyhook control compared to passive control near the shaft system’s resonance point.

Similarly, by employing an acceleration value of $\alpha =7.5\pi \;rad/s^{2}$, the effects of skyhook control and passive control on the shaft system’s torsional vibration were compared using specific data, as shown in Figure 17. The findings reveal reductions of 15.8% and 24.1%, respectively, in the amplitudes of the main shaft’s torsional vibration fluctuations and angular acceleration fluctuations under skyhook control near the shaft system’s resonance point when compared to passive control.

Furthermore, an acceleration value of $\alpha =5\pi \;rad/s^{2}$ was selected to investigate the effects of skyhook control and passive control on the shaft system’s torsional vibration, as illustrated in Figure 18. The results demonstrate that near the shaft system’s resonance point, skyhook control resulted in reductions of 20.0% and 22.8%, respectively, in the amplitudes of the main shaft’s torsional vibration fluctuations and angular acceleration fluctuations, compared to passive control. The MRTVD coil current changes in real time, as shown in the following Figure 19.

By observing Figure 16, Figure 17 and Figure 18, the following conclusions can be drawn: As the acceleration decreases, the time during which the main shaft operates within the resonance range gradually prolongs. However, an encouraging aspect is that remarkable stability in suppressing the main shaft’s torsional vibration is exhibited by the skyhook algorithm, successfully reducing the amplitudes of angular velocity and angular acceleration of the shaft system near the resonance range. This implies that regardless of the changes in resonance duration, the shaft vibration can be effectively controlled by the skyhook algorithm, providing reliable assurance for the stable operation of the shaft system.

Figure 20 illustrates the process of the main shaft rising from rest to the operating speed under different accelerations and compares the inhibitory effect of the MRTVD on the main shaft’s torsional vibration under skyhook control and passive control. The experimental results demonstrate that, under skyhook control, MRTVD exhibits stable and efficient performance in suppressing the torsional vibration of shaft systems. Specifically, the amplitude of the torsional vibration angle of the main shaft decreases by more than 15%, and the amplitude of the torsional vibration angle acceleration of the main shaft decreases by more than 22%. These data further validate the significant inhibitory effect of MRTVD under skyhook control on the main shaft’s torsional vibration. In conclusion, the experimental results underscore that MRTVD, under skyhook control, can effectively and stably suppress the torsional vibration of the shaft system, providing essential assurance for the reliable operation of the main shaft.

## 5. Conclusions

An innovative MRTVD is proposed in this study, and an experimental apparatus for a torsional vibration system is established, which includes the MRTVD. Several experiments are conducted on the experimental setup to investigate the shaft output response in the presence of MRTVD, aiming to improve the stability and smoothness of the shaft system and validate the effectiveness of MRTVD in torsional vibration control. Based on these experiments, the following significant conclusions are drawn:(1)The design and manufacturing of MRTVD are carried out, and its damping principle is investigated. The study reveals that the damping ratio and frequency ratio of MRTVD have a notable impact on the dynamic amplification factor of shaft vibration. At any given excitation frequency, there exists an optimal dynamic damping value that minimizes the torsional vibration amplitude of the main system.(2)The dynamic model of MRTVD is derived, and a torsional vibration dynamic model based on MRTVD is established. The parameters of the torsional vibration system are determined via theoretical calculations. The relationship between the optimal damping and the resonance frequency and inertia ratio of the shaft system is obtained via theoretical analysis. The research demonstrates that during the operation of the shaft system, the optimal damping control effect of MRTVD can be achieved by adjusting the excitation current of its coil according to various resonance frequencies and different operating speeds of the shaft. While the Bingham model has a simple structure, few variables, and clear physical significance, enabling it to effectively represent the damping force–displacement response, its performance in capturing the nonlinearity of the damping force–velocity response is inadequate. The fitting performance of the nonlinear hysteresis characteristics of the damping force–velocity curve is poor, and its nonlinear lag characteristics cannot be well represented. Therefore, subsequent modifications will be made to the dynamic model to better match semi-active control.(3)The experimental phase involved the implementation of semi-active control experiments, followed by an extensive analysis of the torsional vibration system. The results reveal that variations in the current lead to a substantial reduction in the torsional amplitude of the spindle system during resonance, specifically within a discernible range. Importantly, the identification of an optimal current emphasizes its effectiveness in minimizing the torsional vibration of the shaft system. In consideration of real-world operational scenarios, the skyhook control method is employed to assess the efficacy of the MRTVD in controlling the torsional vibration of the main shaft system. Experimental outcomes demonstrate that, during the main shaft’s passage through critical speed at varying accelerations, there is a reduction exceeding 15% in the amplitude of the shaft’s torsional vibration and over 22% in the amplitude of the shaft’s torsional angular acceleration. These results serve to further substantiate the notable inhibitory impact of the MRTVD on spindle torsional vibration when subjected to overhead hook control. The obtained research outcomes establish a robust groundwork for subsequent stages involving actual vehicle experiments, anticipating an extension of the application of magnetorheological technology in addressing torsional vibrations within transmission shaft systems.

## Figures and Tables

**Figure 1 micromachines-15-00236-f001:**
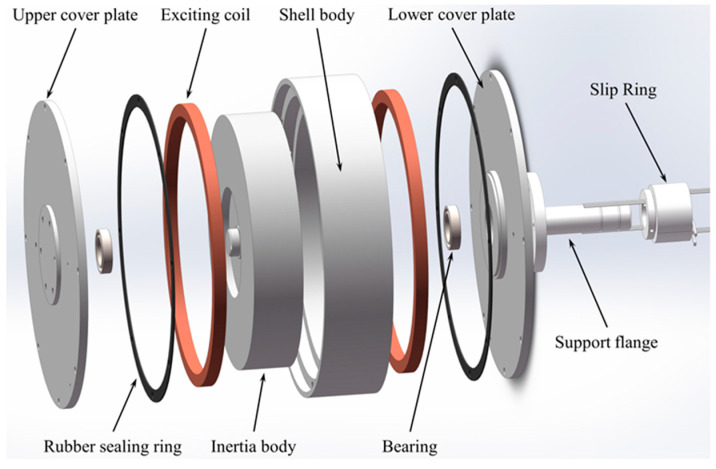
Structure of MRTVD.

**Figure 2 micromachines-15-00236-f002:**
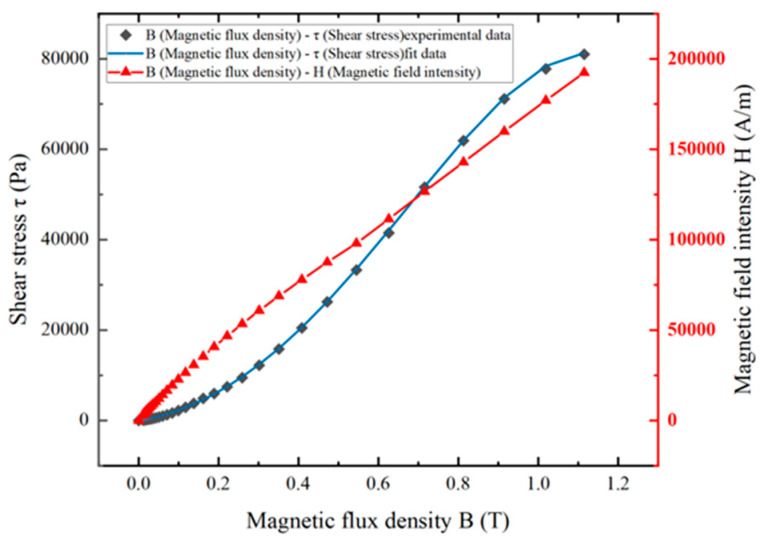
Magnetization characteristics and mechanical properties of MRF.

**Figure 3 micromachines-15-00236-f003:**
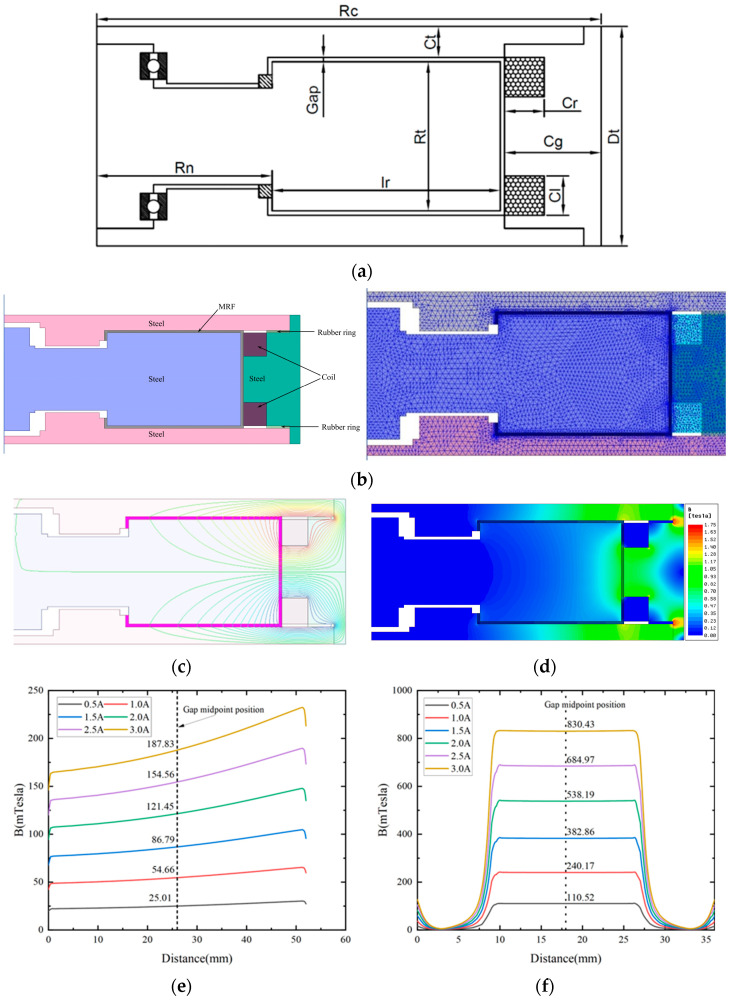
Magnetic field simulation analysis. (**a**) MRTVD model. (**b**) Boundary conditions and finite element mesh partitioning. (**c**) Direction of magnetic field lines. (**d**) Magnetic induction intensity distribution. (**e**) End face gap magnetic field strength. (**f**) Circumferential gap magnetic field strength.

**Figure 4 micromachines-15-00236-f004:**
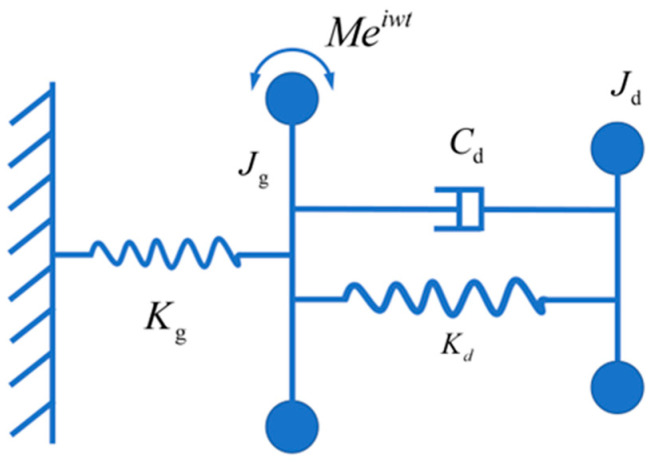
Equivalent model of shaft system.

**Figure 5 micromachines-15-00236-f005:**
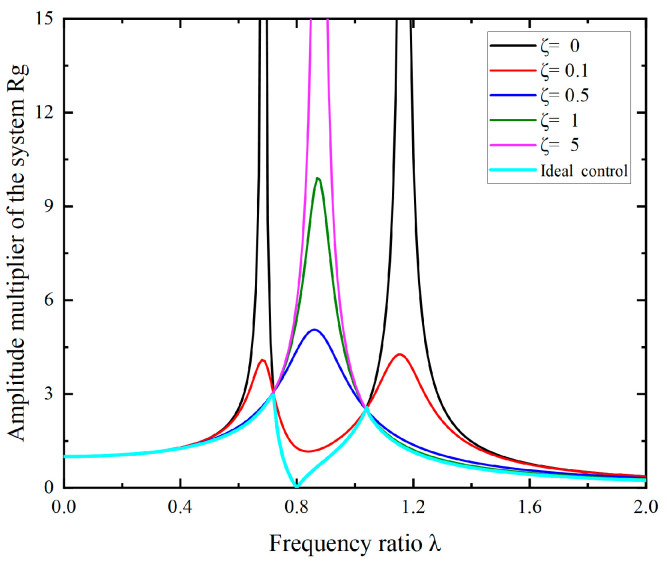
Effect of damping parameters on torsional vibration amplitude of shafting system.

**Figure 6 micromachines-15-00236-f006:**
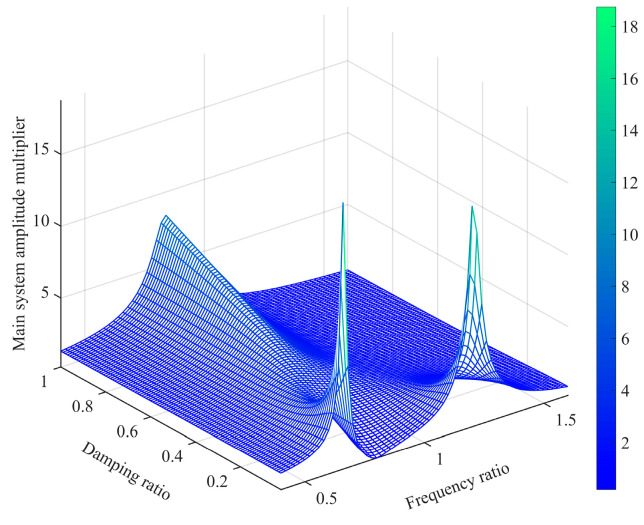
Torsional amplitude amplification factor surface.

**Figure 7 micromachines-15-00236-f007:**
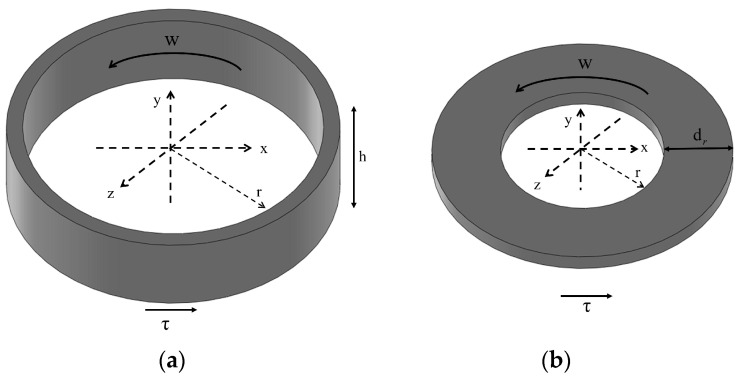
MRF force unit: (**a**) the annular fluid element on the circumferential surface; (**b**) the annular fluid element on the end face.

**Figure 8 micromachines-15-00236-f008:**
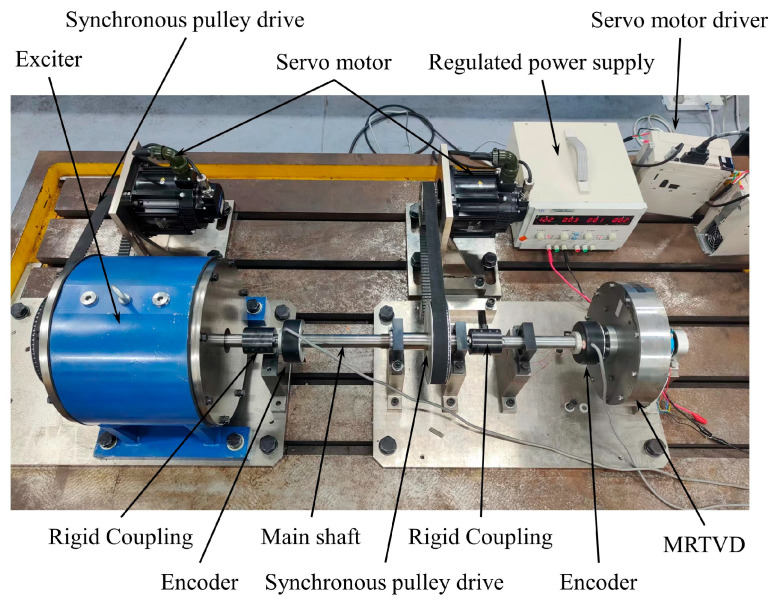
Experimental setup of shaft torsional vibration system.

**Figure 9 micromachines-15-00236-f009:**
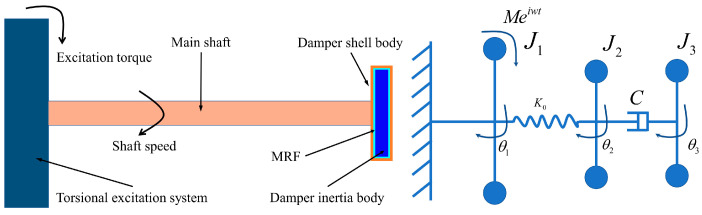
Structure and equivalent model of torsional vibration system.

**Figure 10 micromachines-15-00236-f010:**
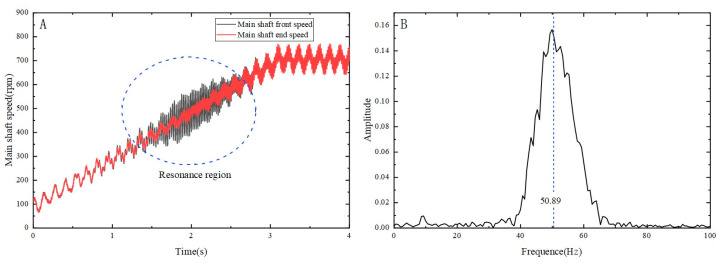
Shafting torsional vibration signal: (**A**) main shaft speed fluctuation; (**B**) Fourier analysis.

**Figure 11 micromachines-15-00236-f011:**
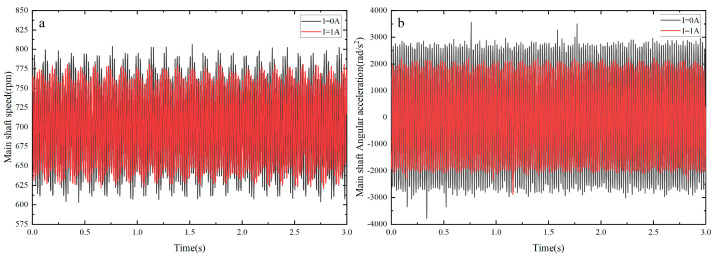
Torsional vibration with different current controls: (**a**) Main shaft speed fluctuation; (**b**) main shaft angular acceleration fluctuation.

**Figure 12 micromachines-15-00236-f012:**
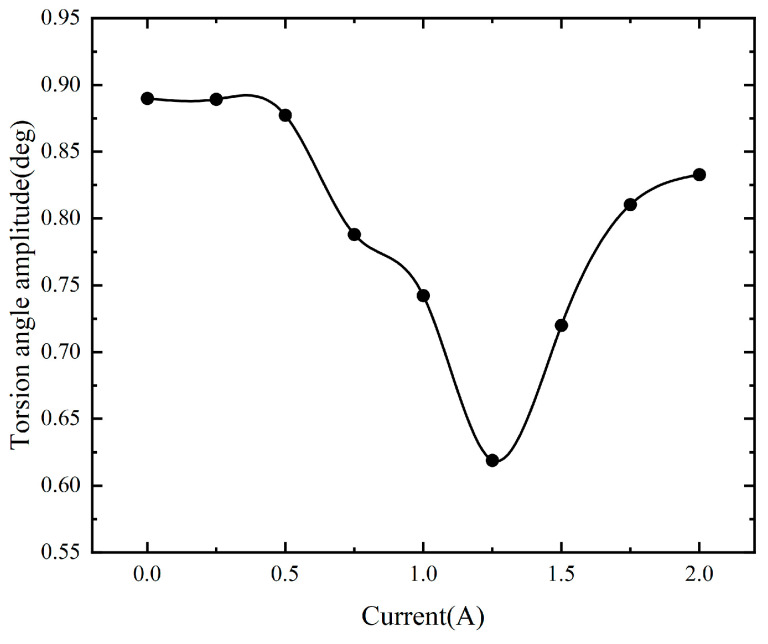
Torsional vibration response under different excitation currents.

**Figure 13 micromachines-15-00236-f013:**
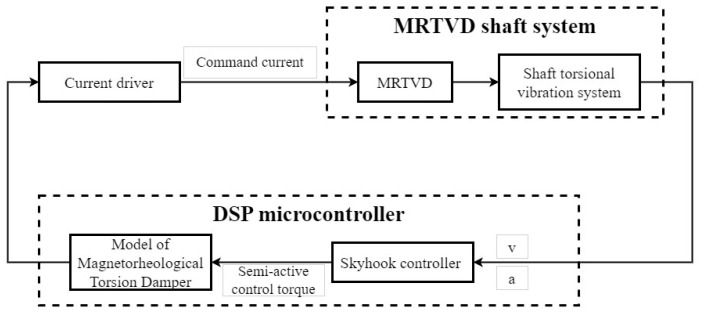
MRTVD shaft system with controller.

**Figure 14 micromachines-15-00236-f014:**
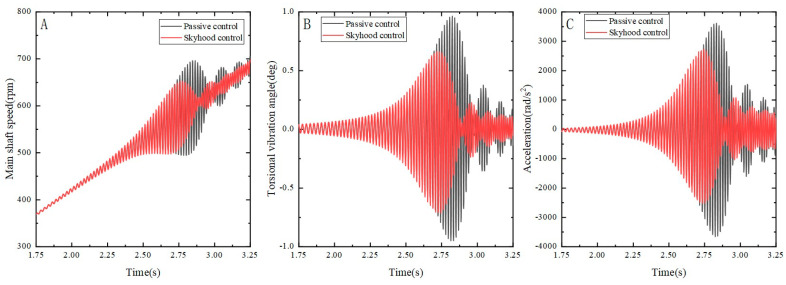
Torsional vibration of the shaft system under different control methods. (**A**) Main shaft speed fluctuation; (**B**) torsional vibration angle fluctuation; (**C**) Main shaft angular acceleration fluctuation.

**Figure 15 micromachines-15-00236-f015:**
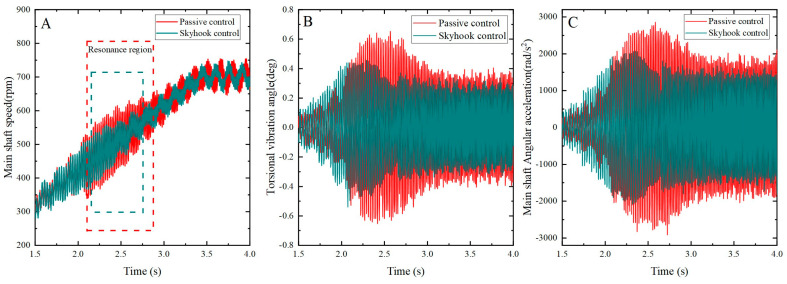
Torsional vibration of the shaft system under different control methods. (**A**) Main shaft speed fluctuation; (**B**) torsional vibration angle fluctuation; (**C**) main shaft angular acceleration fluctuation.

**Figure 16 micromachines-15-00236-f016:**
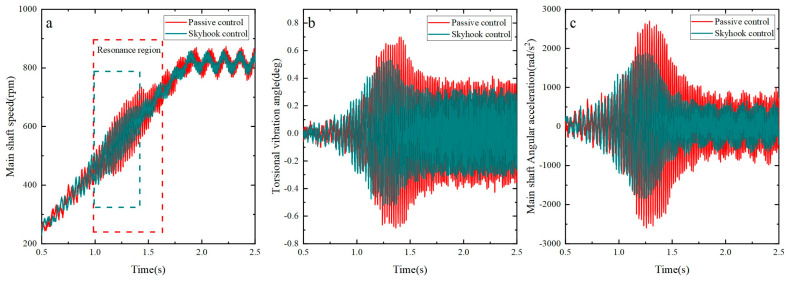
Torsional vibration of the shaft system under different control methods. (**a**) Main shaft speed fluctuation; (**b**) torsional vibration angle fluctuation; (**c**) main shaft angular acceleration fluctuation.

**Figure 17 micromachines-15-00236-f017:**
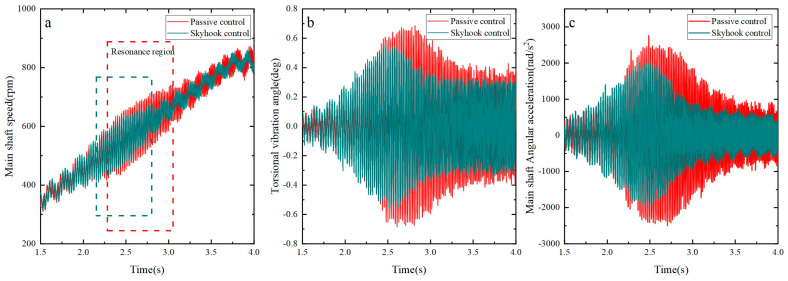
Torsional vibration of the shaft system under different control methods. (**a**) Main shaft speed fluctuation; (**b**) torsional vibration angle fluctuation; (**c**) main shaft angular acceleration fluctuation.

**Figure 18 micromachines-15-00236-f018:**
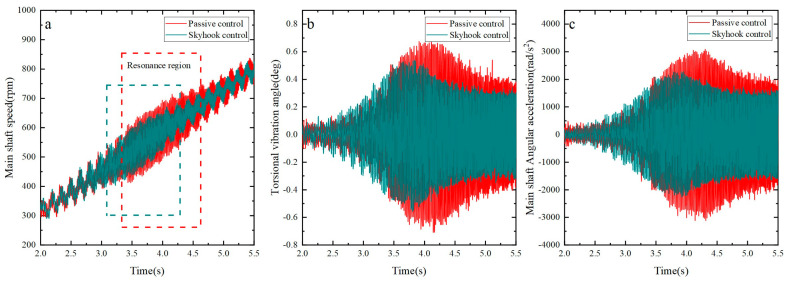
Torsional vibration of the shaft system under different control methods. (**a**) Main shaft speed fluctuation; (**b**) torsional vibration angle fluctuation; (**c**) main shaft angular acceleration fluctuation.

**Figure 19 micromachines-15-00236-f019:**
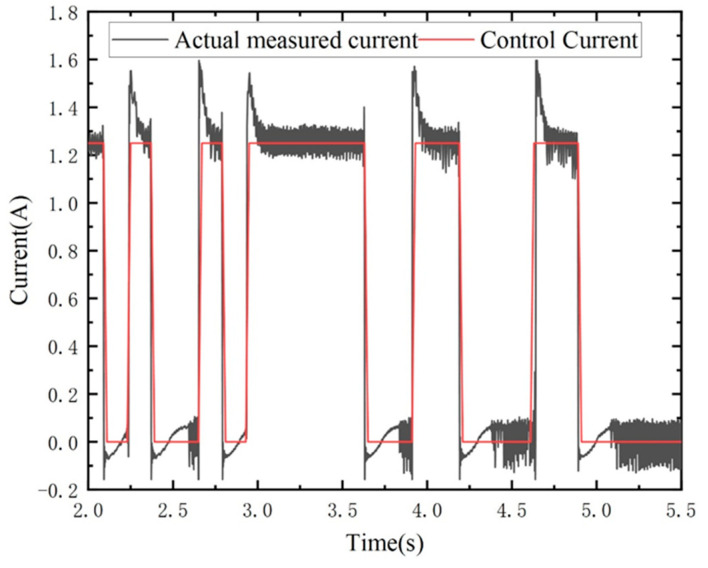
Real-time variation in MRTVD current.

**Figure 20 micromachines-15-00236-f020:**
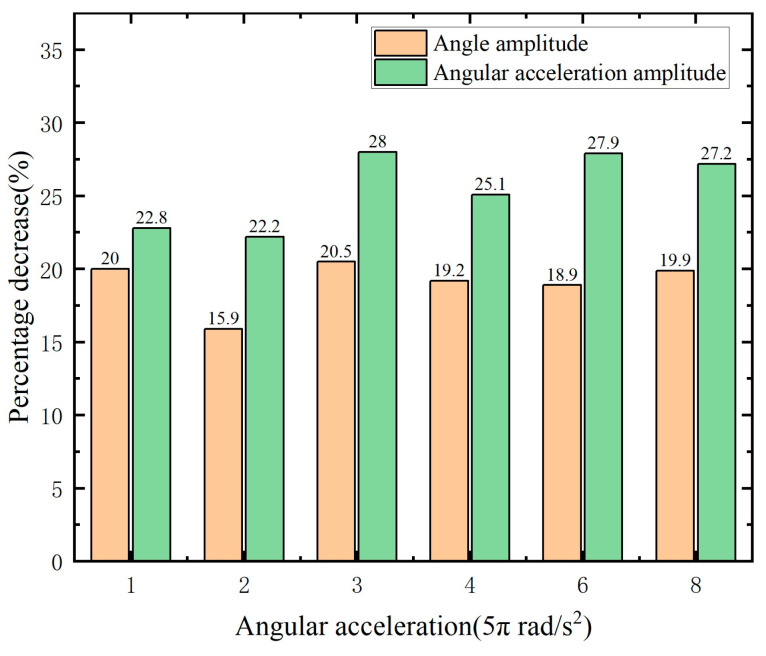
Decrease in skyhook control compared to passive control.

**Table 1 micromachines-15-00236-t001:** Relevant fluid parameters of MRF-350.

Parameters	Value	Unit
Fluid zero-field viscosity ($\eta_{0}$)	12.6	Pa·s
Yield viscosity ($\eta_{d}$)	14.41	Pa·s
Zero-field shear stress ($\tau_{0}$)	126	Pa
Shear yield stress ($\tau_{\infty }$)	85,000	Pa

**Table 2 micromachines-15-00236-t002:** Structural parameters of MRTVD.

Structure Parameters	Value/mm
Rn (radius of inner circle of inertia ring)	40
Ir (radial dimension of inertia ring)	52
Cg (shell thickness)	22
Ct (thickness of upper and lower plates)	6
Rt (thickness of inertia ring)	36
Cr (coil thickness)	9
Cd coil diameter	0.8

**Table 3 micromachines-15-00236-t003:** Parameters of torsional vibration system.

Parameter	Value	Parameter	Value
$J_{1}$ (kg·m^2^)	0.214	$J_{3}$ (kg·m^2^)	0.016
$J_{2}$ (kg·m^2^)	0.029	$K_{0}$ (Nm/rad)	4031

**Table 4 micromachines-15-00236-t004:** Peak and RMS values of torsional vibration response.

Control Methods	Peak of *e* (deg)	Peak of *e* (rad/s^2^)	RMS Value of *e* (deg)	RMS Value of *e* (rad/s^2^)
Passive control	0.9641	3654.8	0.22	812.57
Skyhook control	0.7117 (26.18%)	2730.7 (25.28%)	0.17 (22.73%)	609.3 (25.02%)

**Table 5 micromachines-15-00236-t005:** Peak and RMS values of torsional vibration response.

Control Methods	Peak of *e* (deg)	Peak of *e* (rad/s^2^)	RMS Value of *e* (deg)	RMS Value of *e* (rad/s^2^)
Passive control	0.65	2912.5	0.278	1301.6
Skyhook control	0.54 (16.9%)	2085.1 (28.4%)	0.223 (19.8%)	1039.7 (20.1%)

## Data Availability

The original contributions presented in the study are included in the article, and further inquiries can be directed to the corresponding author.

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
