# Peer review of "Experimental Study on the Skyhook Control of a Magnetorheological Torsional Vibration Damper"

_micromachines, 2024, doi:10.3390/mi15020236_

Round 1
Reviewer 1 Report
Comments and Suggestions for Authors
The authors reported MRTVD construction, theoretical model and experimental results for it. The work is well written and organized. Publication could be considered if the authors carefully address the following comments:
1. Page 2, line 59. It is written that “Numerous scholars have conducted relevant research and exploration on magnetorheological torsional vibration dampers”. Nevertheless, authors give just 4 references. It would be great if the authors added examples of other magnetorheological torsional vibration dampers.
2. It is necessary to indicate the materials from which the damper elements are made.
3. In my opinion Figure 2 should be divided into 2. It is also necessary to indicate how the given curves were measured.
4. Page 9, line 281. It is written that “the excitation frequency was incrementally and uniformly raised, commencing from 0 Hz”. It is not clear, what is frequency 0 Hz?
5. It is not clear, how results shown in Figure 10 were obtained.
6. Why is the second mode of oscillation at a frequency of 50.9 Hz more pronounced than the first at a frequency of 11.8 Hz?
7. Figure 12 shows the existence of some optimal current. It is advisable to give reasons in the text why a certain meaning exists.
8. It is advisable to provide the errors in the results obtained.
Reviewer 2 Report
Comments and Suggestions for Authors
